# Purely antiferromagnetic magnetoelectric random access memory

Tobias Kosub[1,2], Martin Kopte[1,2], Ruben Hühne[3], Patrick Appel[4], Brendan Shields[4], Patrick Maletinsky[4], René Hübner[2], Maciej Oskar Liedke[5], Jürgen Fassbender[2], Oliver G. Schmidt[1] & Denys Makarov[1,2]

Magnetic random access memory schemes employing magnetoelectric coupling to write binary information promise outstanding energy efficiency. We propose and demonstrate a purely antiferromagnetic magnetoelectric random access memory (AF-MERAM) that offers a remarkable 50-fold reduction of the writing threshold compared with ferromagnet-based counterparts, is robust against magnetic disturbances and exhibits no ferromagnetic hysteresis losses. Using the magnetoelectric antiferromagnet $Cr_2O_3$, we demonstrate reliable isothermal switching via gate voltage pulses and all-electric readout at room temperature. As no ferromagnetic component is present in the system, the writing magnetic field does not need to be pulsed for readout, allowing permanent magnets to be used. Based on our prototypes, we construct a comprehensive model of the magnetoelectric selection mechanisms in thin films of magnetoelectric antiferromagnets, revealing misfit induced ferrimagnetism as an important factor. Beyond memory applications, the AF-MERAM concept introduces a general all-electric interface for antiferromagnets and should find wide applicability in antiferromagnetic spintronics.

[1] Institute for Integrative Nanosciences, Institute for Solid State and Materials Research (IFW Dresden e.V.), 01069 Dresden, Germany. [2] Helmholtz-Zentrum Dresden-Rossendorf e.V., Institute of Ion Beam Physics and Materials Research, 01328 Dresden, Germany. [3] Institute for Metallic Materials, Institute for Solid State and Materials Research (IFW Dresden e.V.), 01069 Dresden. [4] Department of Physics, University of Basel, 4056 Basel, Switzerland. [5] Helmholtz-Zentrum Dresden-Rossendorf e.V., Institute of Radiation Physics, 01328 Dresden, Germany. Correspondence and requests for materials should be addressed to T.K. (email: t.kosub@hzdr.de) or to D.M. (email: d.makarov@hzdr.de).

In the effort to develop low-power data processing and storage devices, nonvolatile random access memory schemes have received considerable attention[1]. Magnetic elements such as the magnetic random access memory (MRAM) (Fig. 1a) promise excellent speed, superior rewritability and small footprints, which has led to strong commercial interest in this technology for memory applications. In addition to ferromagnetic MRAM, two complementary approaches have recently emerged for advancing beyond conventional MRAM elements in terms of its writing power and data robustness. On the one hand, switching and reading the antiferromagnetic order parameter of metallic antiferromagnets with charge currents[2,3] has enabled purely antiferromagnetic MRAM (AF-MRAM), granting superior data stability against large magnetic disturbances and potentially even faster switchability. On the other hand, magnetoelectric random access memory (MERAM) promises energy efficient writing of antiferromagnets, by eliminating the need for charge currents through the memory cell and instead relying on electric field-induced writing. Reading out the antiferromagnetic state from MERAM has presented a challenge to date as magnetoelectric antiferromagnets (for example, $BiFeO_3$ or $Cr_2O_3$) are dielectrics. Therefore, the readout signal of MERAM cells is conventionally acquired from a ferromagnet that is coupled with the magneto-electric antiferromagnet by exchange bias[4–8]. While ferromagnets enable readability, their presence strongly interferes with the magnetoelectric selection of the antiferromagnetic order parameter[9]. This is related to exchange bias and ferromagnetic hysteresis, both of which need to be overcome in the writing process of MERAM with ferromagnets.

Here we put forth the concept of purely antiferromagnetic MERAM (AF-MERAM) (Fig. 1a), which avoids the issues associated with the presence of ferromagnets by instead using polarizable paramagnets, for example, Pt, to probe the order parameter of the magnetoelectric antiferromagnet. As shown schematically in Fig. 1b, the prototypical memory cell consists of an active layer of insulating magnetoelectric antiferromagnet, a bottom gate electrode for writing purposes and a top electrode that provides the readout interface via anomalous Hall measurements[10]. Using $Cr_2O_3$ as an AF element, we demonstrate a complete working AF-MERAM cell, proving that this concept yields substantial improvements in terms of magnetoelectric performance over comparable MERAM realizations with ferromagnets. In particular, by removing the ferromagnetic component from MERAM, we reduce the writing threshold by a factor of about 50. These characteristics render AF-MERAM a promising new member to the emerging field of purely antiferromagnetic spintronics[3,11]. We show the magnetoelectric writing and all-electric reading operations of a cell at room temperature over hundreds of read–write cycles. While nonvolatile solid-state memory is one possible application of AF-MERAM cells, the concept is applicable to other fields of antiferromagnetic spintronics, such as logics, magnonics[12] and material characterization.

## Results

**Room temperature operation of AF-MERAM.** To realize the memory cell, we use an epitaxial layer stack of Pt(20 nm)/ α-$Cr_2O_3$(200 nm)/Pt(2.5 nm) that is prepared on $Al_2O_3$(0001) substrates. Similar stacks with α-$Cr_2O_3$ have been extensively studied in the scope of traditional MERAM elements with ferromagnetic Co layers[4,5,13–15]. The thicker bottom Pt film serves as the gate electrode and the thin Pt top layer is used to measure the AF order parameter all-electrically via zero-offset anomalous Hall magnetometry[10] (hereafter zero-offset Hall). This readout approach makes use of the uncompensated boundary magnetization of α-$Cr_2O_3$(0001), which is rigidly coupled to the AF bulk and creates proximity magnetization in the Pt film[16,17].

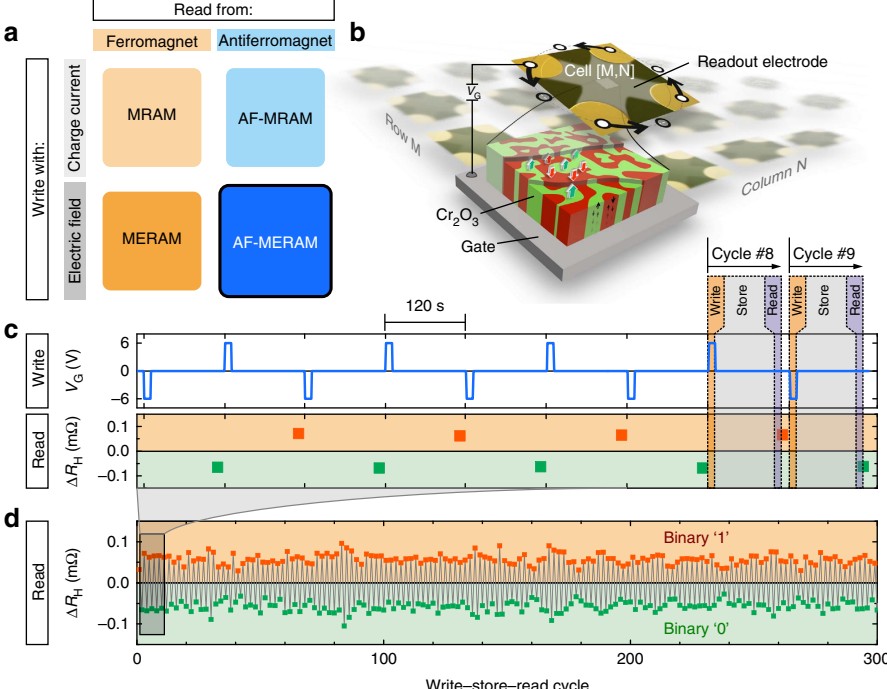

**Figure 1 | Electric field-driven manipulation of the antiferromagnetic order parameter.** (**a**) Nonvolatile magnetic random access memory elements categorized according to their writing and readout interfaces. Antiferromagnetic magnetoelectric random access memory (AF-MERAM) initiates a new field of antiferromagnetic spintronics. (**b**) Sketch of one memory cell within a matrix of devices. The arrows indicate the contact permutation to obtain offset free Hall readings[10]. (**c**) Random access memory operation where binary information is written by a voltage pulse and stored in the antiferromagnetic order parameter. The magnetic state is readout at a later time after the writing stimulus is removed. (**d**) Device behavior over 300 write–store–read cycles.

An individual magnetoelectric element is obtained by patterning the top Pt layer. Figure 1c shows the protocol of an isothermal magnetoelectric switching experiment that was carried out at 19 °C in a permanent magnetic field of $H \approx +0.5\,MA\,m^{-1}$ along the film normal. The test sequence mimics random access operations comprising the three essential elements of any memory cell: writing, storage and reading. One of the key technological advantages is that the memory cell operates in static magnetic fields and writing operations are triggered by the application of a voltage. No energy input is necessary during the storage times. The reproducibility of this process is demonstrated over 300 write–store–read cycles in Fig. 1d, during which the cell reveals no performance degradation.

Two key material requirements must be satisfied to achieve reliable magnetoelectric reversal processes such as shown in Fig. 1. First, the order parameter has to be susceptible to the gate voltage via the linear magnetoelectric effect. Second, the cell has to exhibit thermal stability at the operation temperature, giving rise to stable remanent magnetic states. Both criteria can be directly probed in our system using the electrical writing and reading interfaces of the magnetoelectric cell. To reveal the exact influence of magnetic and electric field on the antiferromagnetic order parameter, it is mandatory to avoid the influence of magnetic anisotropy, which fixes the order parameter while below the ordering temperature, and instead carry out magnetoelectric field cooling through the ordering temperature. The map in Fig. 2a shows the resulting average antiferromagnetic order parameter in the cell after cooling from 30 to 7 °C using the indicated combination of magnetic cooling field $H_{cool}$ and electric cooling field $E_{cool} = V_{cool}/t$ ($t$ denotes the AF film thickness). For large $EH$ fields, the order parameter selection is consistent with that expected in $\alpha$-$Cr_2O_3$ (refs 7,13,18,19) due to the linear magnetoelectric effect. However, for small writing voltages that are technologically desirable, the $EH$ symmetry is disturbed, giving rise to magnetic field-induced selection of the order parameter. Strikingly, the $EH$ symmetry is perfectly restored when accounting for a gate bias voltage $V_{GB}$, which is about $-1\,V$ for this system.

When applying a writing voltage to the cell at 19 °C, the antiferromagnetic order parameter can be switched hysteretically with a coercive gate voltage $V_C$ of $\approx 1.5\,V$ (Fig. 2b), completing the list of ingredients for the nonvolatile AF-MERAM prototype. The slightly asymmetric shape of the hysteresis loop is due to the gate voltage range being symmetric about $V_G = 0$, instead of $V_G = V_{GB}$. The temperature window, in which magnetoelectric writing can be carried out, is limited at higher temperatures by the collapse of antiferromagnetic order and at lower temperatures by magnetic anisotropy[15]. It should be possible to widen this writability window considerably to $>100\,K$. The high-temperature limit can be enhanced by doping[20,21], and the lower-temperature limit by applying higher writing voltages[4,5] or by intentionally reducing the anisotropy via doping[20] (Supplementary Note 1).

Table 1 contains an overview of state-of-the-art studies of magnetoelectric functionality using magnetoelectric thin films[4–6,9,13] and single crystals[7], but in both cases relying on interfacial exchange bias with a thin ferromagnetic layer. In addition, the AF-MERAM cell presented in this work is included for comparison. The metrics in the overview are the magnetoelectric film thickness $t$, the writing threshold $(VH)_C$ and the coercive gate voltage $V_C$. For integration in microelectronics, the latter two are of particular relevance as the circuit voltage rating depends on them.

While exchange bias has traditionally been used to probe the antiferromagnetic state of $Cr_2O_3$, this leads to strongly increased magnetoelectric coercivities, especially for thin films of $Cr_2O_3$ (refs 4,5). When judging the writing threshold, all the exchange bias systems require very large $VH$ for isothermal magnetoelectric switching of the AF order parameter in $Cr_2O_3$. In contrast, the AF-MERAM approach provides a route to reduce both the coercivity and the resulting write voltage by a factor of about 50 over exchange-biased examples (Supplementary Note 2). In addition, AF-MERAM can be readout in permanent external magnetic fields, whereas exchange-biased MERAM requires the removal of the magnetic field for readout. Thus, the example here presented opens an appealing field of AF-MERAM with ultra-low

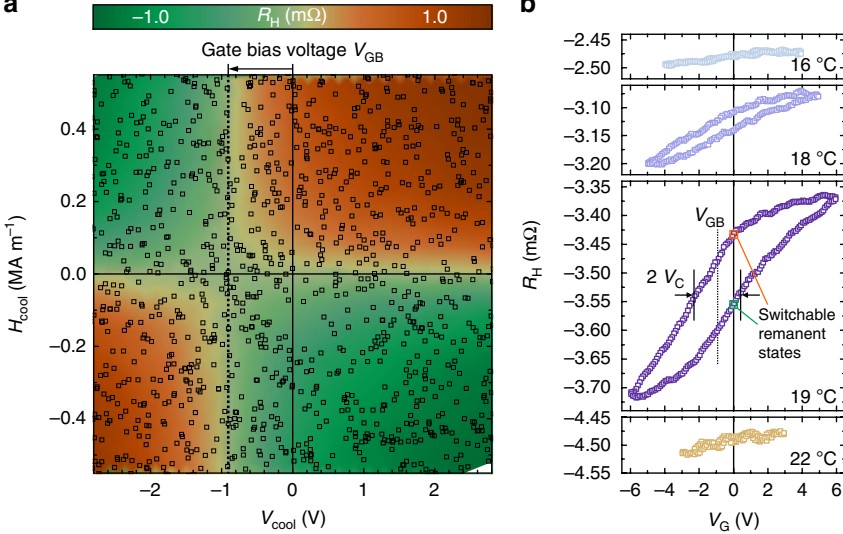

**Figure 2 | Isothermal and field-cooled magnetoelectric selection.** (**a**) Map of the antiferromagnetic state selected by a range of magnetic field and gate voltage combinations during cool-down from 30 °C through the antiferromagnetic ordering temperature to the measurement temperature of 7 °C. Measurements were carried out at $H = 0$ and $V_G = 0$. The squares are data points and the background color is a linear interpolation. (**b**) Readout signal corresponding to the antiferromagnetic order parameter of the cell as a function of the writing voltage $V_G$ for several temperatures near the antiferromagnetic ordering temperature and $H = 0.5\,MA\,m^{-1}$. The open hysteresis loop with coercivity $V_C$ gives rise to switchable remanent states.

**Table 1 | Performance chart of MERAM systems.**

| Study | $t$ (μm) | $(VH)_C$ (MW m$^{-1}$) | $V_C$ (V) | Magnetic field |
|---|---|---|---|---|
| Exchange bias reversal*, $Cr_2O_3$/Co/Pd (ref. 7) | 1,000 | 240 | 450[†] | Writing pulse |
| Exchange bias reversal*, $Cr_2O_3$/Co/Pt (ref. 5) | 0.2 | 40 | 56[†] | Writing pulse |
| Exchange bias reversal*, $Cr_2O_3$/Co/Pt (ref. 4) | 0.5 | 48 | 105[†] | Writing pulse |
| Magnetization switching, $BiFeO_3$/CoFe (ref. 6) | 0.1 | — | 4 | Must be $\approx 0$ for readout |
| Magnetization switching, $Cr_2O_3$/Pt (present work) | 0.2 | 0.75 | 1.5 | Permanent |

Overview of state-of-the-art isothermal magnetoelectric switching studies using either the linear magnetoelectric effect in $Cr_2O_3$ or the multiferroic coupling in $BiFeO_3$. The value $(VH)_C$ gives the magnetoelectric writing threshold (product of magnetic field and voltage). The writing voltage $V_C$ allows to qualitatively compare $Cr_2O_3$-based systems and $BiFeO_3$ systems in terms of the voltage at which the magnetization state switches.
*Application of the writing voltage does not switch the ferromagnetic Co, but only the antiferromagnetic $Cr_2O_3$, implying that the magnetic field must be removed for readout from the ferromagnet.
†For comparability, the writing voltages are calculated for a magnetic field of $H_{write} = 0.5\,MA\,m^{-1}$ as was used in the present study. The actual used writing voltages in these studies are similar to the normalized values, as the magnetic fields were also similar.

writing thresholds and superior stability and readability of the magnetic information in the presence of external magnetic fields.

## Discussion

The gate bias voltage of $V_{GB} \approx -1V$ (Fig. 2) presents a key challenge for achieving ultra-low voltage threshold switching and ultra-high data stability. It has a detrimental effect on both the required writing voltage and on the data stability at zero voltage as the antiferromagnetic state develops a susceptibility to magnetic fields, even in the absence of an electric field ($V_G = 0$). We find that the gate bias voltage is a material characteristic in thin films of magnetoelectric antiferromagnets, and in the following we reveal its physical origin and derive a means to control its value.

When combining the large body of data on $Cr_2O_3$ thin-film systems[4,5,7,9,13,15,17,22–29], a coherent picture emerges: the total magnetoelectric energy density exerting a selection pressure on the antiferromagnetic order parameter in thin-film magnetoelectric antiferromagnets is composed of three effects that act simultaneously:

$$U_{MEAF}(E, H) = t^{-1}\alpha V_G H + t^{-1} J_{EB} + t^{-1}\rho_m \mu_0 H \qquad (1)$$

The first term describes the linear magnetoelectric effect with its coefficient $\alpha$ reported to be about $1\,ps\,m^{-1}$ in $Cr_2O_3$ (refs 18,30). This is the only desired effect in the context of MERAM devices, while the other two effects are parasitic. The second term is the influence of the exchange bias coupling strength $J_{EB}$ on the antiferromagnet. While this term was typically the strongest contribution in previous studies (Supplementary Note 3), it is zero in AF-MERAM due to the lack of a ferromagnet. The last term arises from a non-zero areal magnetic moment density $\rho_m$ within the antiferromagnet itself, which renders the material ferrimagnetic. This term, due to emergent ferrimagnetism, cannot be excluded a priori. The gate bias voltage can now be calculated from equation (1):

$$V_{GB} = -\mu_0 \frac{\rho_m}{\alpha} \qquad (2)$$

The gate bias voltage $V_{GB}$ is in an intimate relation with the magnetoelectric coefficient $\alpha$ and the areal magnetic moment density $\rho_m$ at the onset temperature of the thermal stability of the antiferromagnetic order. Equation (2) implies that the gate bias in magnetoelectric field cooling experiments vanishes for perfectly antiferromagnetic order ($\rho_m = 0$). Its non-zero value in our system can be used to estimate the approximate ferrimagnetic moment density at the ordering temperature of about $21\,°C$, yielding a value of $\rho_m \approx 0.1\mu_B nm^{-2}$. Conversely, achieving a low gate bias voltage requires that ferrimagnetism is averted.

The presence of ferrimagnetism cannot be accounted for by any intrinsic effect within the $Cr_2O_3$ antiferromagnetic film, as all

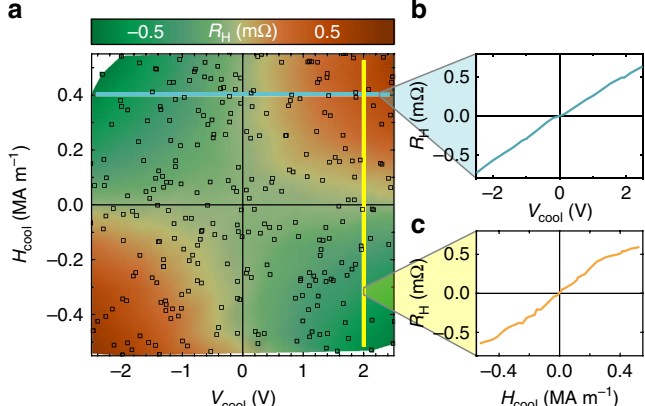

**Figure 3 | Influence of a $V_2O_3$ buffer layer on $Cr_2O_3$ magnetoelectricity.** (**a**) Magnetoelectric field cooling map of the antiferromagnetic order selection for a $V_2O_3$/$Cr_2O_3$/Pt system. (**b,c**) Line profiles taken from the map in **a**. Only non-zero products of gate voltage and magnetic field lead to appreciable order parameter selection, whereas the individual stimuli do not.

magnetic moments, including boundary moments[10,16,17], are intrinsically compensated when accounting for all boundaries (Supplementary Note 4; Supplementary Fig. 1). Therefore, extrinsic effects are necessary to break the sublattice equivalence and produce ferrimagnetism. In the following, we present an in-depth study of extrinsic thin-film phenomena and their influence on the emergent ferrimagnetism. Namely, we invoke different degrees of crystalline twinning, elastic lattice deformation, intermixing and misfit dislocation density in $Cr_2O_3$ thin films, by preparing three distinct systems with epitaxial underlayers of $Al_2O_3$(0001), Pt(111) or $V_2O_3$(0001).

One striking result is that the gate bias voltage, and thus the emergent ferrimagnetism, can indeed be controlled by the choice of underlayer material. In particular, when $Cr_2O_3$ thin films are grown on a $V_2O_3$ underlayer, ferrimagnetism is almost entirely eliminated. Figure 3a shows a magnetoelectric field cooling map of the $V_2O_3$-buffered system exhibiting virtually perfect $EH$ symmetry. As highlighted by the indicated line profiles (Fig. 3b,c), the selection preference for a particular antiferromagnetic state vanishes when either $E = 0$ or $H = 0$, as expected from the pristine action of the linear magnetoelectric effect (first term in equation (1)). The possibility to completely eliminate the gate bias is highly relevant for AF-MERAM applications, as the AF state can then be switched with lower voltages (Supplementary Note 5; Supplementary Fig. 2) and is completely stable once the gate voltage returns to zero.

**Table 2 | Influence of different underlayers on structural and ferrimagnetic properties of Cr$_2$O$_3$ thin films.**

| Underlayer material | Twinning ratio (%) | c axis compression (%) | Expected miscibility | Linear misfit (%) | Areal magnetization $\rho_m$ (a.u.) |
|---|---|---|---|---|---|
| Al$_2$O$_3$ | ≈ 2 | 0.0 | weak | + 4.0 | + 0.455 ± 0.28 |
| Pt | ≈ 50 | 0.18 | none | + 2.8 | + 0.100 ± 0.043 |
| V$_2$O$_3$ | ≈ 2 | 0.30 | strong | − 0.5 | − 0.0021 ± 0.001 |

The values for the structural properties are derived in detail in the (Supplementary Note 6; Supplementary Fig. 3; Supplementary Fig. 4). The ferrimagnetic moment density values are relative values obtained by zero-offset Hall (Supplementary Note 7; Supplementary Fig. 5; Supplementary Fig. 6). They are normalized to the approximate value for Pt/Cr$_2$O$_3$/Pt obtained via the gate bias voltage as of equation (2).

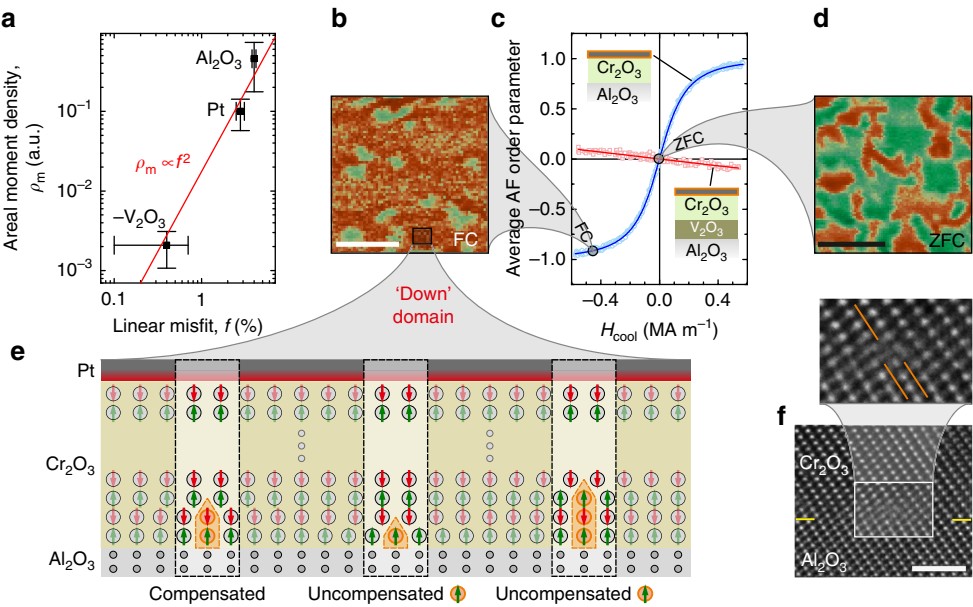

**Figure 4 | Thin-film Cr$_2$O$_3$ behaves ferrimagnetically.** (**a**) Scaling between the measured areal magnetic moment density and the linear misfit between Cr$_2$O$_3$ and its underlayer. Vertical error bars represent s.e.'s based on the best fit values for the domain moment and domain areas (Table 2; Supplementary Note 7). Horizontal error bars show the maximum discrepancy of the lattice misfit values when accounting for only half the thermal expansion of one lattice. (**b,d**) Images of the surface magnetic stray field after field cooling (FC) and zero-field cooling (ZFC), respectively, were obtained by scanning nitrogen-vacancy microscopy (see text). Scale bars, 1 μm. (**c**) The emergent ferrimagnetism couples strongly to external magnetic fields and renders the antiferromagnetic order parameter selectable by magnetic fields much smaller than anisotropy fields. (**e**) Sketch of the effect of different misfit dislocations on the atomic populations of the two antiferromagnetic sublattices. (**f**) High-resolution TEM images of the Al$_2$O$_3$/Cr$_2$O$_3$ interface (yellow guide lines) showing complete structural coherence disrupted by occasional misfit dislocations. Scale bar, 2 nm.

To pinpoint the specific extrinsic effect that is responsible for the different degrees of emergent ferrimagnetism in Cr$_2$O$_3$ thin films grown on different underlayers, it is instructive to correlate the observed areal magnetic moment density and the various growth-induced effects (Table 2). The areal magnetization is determined via the slope of the dependence of the antiferromagnetic order parameter selection on the magnetic field (Fig. 4c; Supplementary Fig. 5). The gradual shape of these dependences results from a selection tendency of uniaxial antiferromagnetic domains according to their ferrimagnetism, averaged over the readout electrode area. To verify that the microscopic ordering is indeed a mixture of purely uniaxial domains, images of the surface magnetization states after zero-field cooling (Fig. 4d) and field cooling (Fig. 4b) were obtained by scanning nitrogen-vacancy (NV) magnetometry[31–33]. This technique measures the stray magnetic field ≈ 50 nm above the sample surface, clearly indicating the equal presence of up- and down-domains in the zero-field-cooled state. In contrast, field cooling predominantly selects one of the two domain orientations that allows to calculate the degree of ferrimagnetism in the films.

It should also be noted that measuring the gate bias voltage $V_{GB}$ (Equation (2)) provides a second route to quantify the areal magnetization $\rho_m$ absolutely, which is however restricted to conducting underlayers and suffers from the uncertainty of the

value of $\alpha$. Therefore, magnetization values determined via the gate bias will not be used for the comparison of the different underlayer materials.

Based on these data, we conclude that elastic film strain, twinning and cation intermixing in epitaxial Cr$_2$O$_3$ films cannot account for the observed degree of ferrimagnetism, as none of these properties are correlated to the areal magnetization (Table 2). Instead, the results suggest that the lattice mismatch is the cause of the emergent ferrimagnetism.

The scaling relationship between the measured areal magnetic moment density and the linear lattice misfit between Cr$_2$O$_3$ and its underlayer is shown in Fig. 4a. When taking into account the data of the three investigated systems, we find that the data align tightly to a quadratic scaling relation (red line). Such a relationship hints at the number of the misfit dislocations per area being the key property determining the areal ferrimagnetic moment density. This result leads to a picture in which the population of the two antiferromagnetic sublattices is unbalanced by the presence of misfit dislocations.

Such dislocations arise due to the heteroepitaxial film growth (Fig. 4f) as the dominant defect type of the otherwise highly coherent interface and appear within the first atomic layers of the Cr$_2$O$_3$ film as evidenced by positron annihilation spectroscopy (Supplementary Note 8; Supplementary Fig. 7; Supplementary

Table 1). As sketched for the case of compressive misfit in Fig. 4e, the dislocations (orange boxes) can contain unequal populations of spin 'up' and spin 'down' atoms if the dislocation terminates after an odd number of atomic layers. These surplus spins are all aligned within one domain due to the layered sublattice structure in $\alpha$-$Cr_2O_3$(0001). While the magnetic moment of atoms near dislocations might be different from atoms in the relaxed lattice, this picture serves to illustrate that misfit dislocations do indeed unbalance the atomic populations in each of the two sublattices.

Remarkably, the lattice misfit not only correlates with the magnitude of the emergent magnetization, but also with its sign with respect to the antiferromagnetic order parameter. This sign change of the ferrimagnetic behavior in the case of tensile misfit for the $V_2O_3$-buffered sample emerges naturally from the previously introduced picture. Tensile misfit results in atoms being skipped from the bottom boundary sublattice instead of atoms being added. Therefore, tensile misfit results in the top boundary magnetization being aligned along the cooling field, while compressive misfit results in the top boundary magnetization being aligned opposite to the magnetic cooling field.

To quantify ferrimagnetism, we investigate the magnetic field-induced antiferromagnetic order parameter selection with no electric field applied (Fig. 4c), which is influenced exclusively by the last term of equation (1). The $Al_2O_3$ buffered films are clearly more susceptible to the magnetic field than the $V_2O_3$-buffered films, which is in line with the substantially larger lattice misfit of the former over the latter. Moreover, a magnetic field of the same sign selects opposite antiferromagnetic states in the two systems, which corresponds to the opposite sign of the lattice misfit.

In conclusion, we demonstrated reliable room temperature magnetoelectric random access memory cells based on a new scheme that relies purely on antiferromagnetic components and does not require a ferromagnet for readout. This AF-MERAM provides substantially reduced writing thresholds over conventional MERAM prototypes, enabling further improvements in the energy efficiency of nonvolatile solid-state memory and logics. Since a permanent magnetic writing field does not interfere with readout in AF-MERAM, this new approach extends voltage driven writing to magnetoelectric antiferromagnets such as $Cr_2O_3$, whereas such functionality has previously been feasible only in multiferroic antiferromagnets such as $BiFeO_3$ (Table 1). It should be noted that the advantages of omitting the ferromagnet from MERAM cells likewise apply to multiferroic antiferromagnets, opening an appealing field of AF-MERAM with ultra-low writing thresholds and superior stability of the magnetic order parameter. The concept also provides an important new building block for the emerging field of antiferromagnetic spintronics. While we did not investigate the speed of the actual writing process, first prototypes of conventional MERAM could be switched within a few tens of nanosecond[5].

We use thin films of magnetoelectric antiferromagnetic $Cr_2O_3$ as the core material and find that this material becomes ferrimagnetic when grown as epitaxial thin films. Emergent ferrimagnetism in thin films of magnetoelectric antiferromagnets can be desirable[34]. For the application to purely antiferromagnetic magnetoelectric elements, however, ferrimagnetism should be minimized. Through an in-depth structural characterization, we find that the observed degree of ferrimagnetism is correlated with the square of the linear lattice misfit between $Cr_2O_3$ and its underlayer. This finding provides both a fundamental mechanism for the phenomenon of emergent ferrimagnetism and suggests a readily available tuning knob to enhance or eliminate the magnetic field coupling of magnetoelectric antiferromagnets.

## Methods

**Sample preparation.** Oxide films were grown by reactive evaporation of the base metal in high vacuum onto $c$-cut sapphire substrates (Crystec GmbH) heated to 700 °C initially and to 500 °C after the first few monolayers. The background gas used was molecular oxygen at a partial pressure of $10^{-5}$ mbar. Chromium was evaporated from a Knudsen cell, vanadium was evaporated from a block target using an electron-beam and platinum was sputtered from a d.c. magnetron source. Deposition of the oxides was carried out using rates of about 0.4 Å s$^{-1}$ and was monitored *in situ* by reflection high-energy electron diffraction. Oxide layers were subjected to a vacuum annealing process at 750 °C and residual pressure of $10^{-7}$ mbar directly after growth. The thin Pt top layers were deposited at lower temperatures of $\approx 100$ °C using a higher rate of 1.0 Å s$^{-1}$ to maintain layer continuity. Hall crosses were patterned from the top Pt layers, by $SF_6$ reactive ion etching around a photoresist mask.

**Transport experiments.** Transport was measured using zero-offset Hall[10]. Typical current amplitudes were on the order of 500 µA. RAM operation was carried out in a permanent magnetic background field of $H \approx +0.5$ MA m$^{-1}$ along the film normal.

To obtain the average AF order parameter dependence on the magnetic cooling field, the data of the spontaneous Hall signal after cooling under a range of field values were fitted by an expression that provides the normalization and the absolute magnetic moment distribution of individual domain pieces within the $Cr_2O_3$ films (Supplementary Note 7). The relative domain sizes of films with different buffer layers were determined using zero-offset Hall by evaluating the statistics of the domain selection within the finite-size Hall crosses (Supplementary Note 7).

The structural properties of the $Cr_2O_3$ films on different buffer layers were characterized by x-ray diffraction and channeling contrast scanning electron microscopy as shown in detail in (Supplementary Note 6).

**NV magnetic microscopy.** Scanning NV magnetometry was performed with a tip fabricated from single-crystal, $<100>$-oriented diamond that was implanted with $^{14}$N ions at 6 keV, and annealed at 800 °C to form NV centres[35]. An external field of 2.2 kA m$^{-1}$ was applied along the NV axis (diamond $<111>$ crystal direction) to induce Zeeman splitting of the NV electronic ground-state spin. A microwave driving field was then locked to the spin transition at $\approx 2.864$ GHz to track the additional Zeeman shift due to the stray field of the magnetic film surface[36]. Magnetic field values for each pixel were obtained by averaging the microwave lock frequency for 7 s.

**Data availability.** The data underlying the present work are available upon request from the corresponding authors.

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

## Acknowledgements

We acknowledge the support from Dr S. Harazim for the maintenance of the clean room facilities, S. Nestler for conducting the reactive ion etching, M. Bauer for depositing the contact pads, R. Engelhard for the maintenance of the deposition tools and D. Karnaushenko and D. D. Karnaushenko for providing polyimide photoresist (all IFW Dresden). Support by the Structural Characterization Facilities at IBC of the HZDR is gratefully acknowledged. This work was funded in part by the European Research Council under the European Union's Seventh Framework Programme (FP7/2007-2013)/ ERC grant agreement no. 306277 and the European Union Future and Emerging Technologies Programme (FET-Open grant no. 618083).

## Author contributions

T.K. prepared the samples. T.K. and M.K. set up the magneto-transport measurement system. T.K. carried out the electrical measurements and the corresponding data analysis. T.K., M.K. and Ru.H. conducted the X-ray diffraction studies. P.A., B.S. and P.M. carried out the NV microscopy measurements. Re.H. imaged cross-sections of the samples in TEM. M.O.L. conducted the positron annihilation spectroscopy (PAS) investigations. T.K. created the graphics and T.K. and D.M. wrote the manuscript with comments from all authors. D.M., O.G.S. and J.F. supervised the project.

## Additional information

**Competing financial interests:** The authors declare no competing financial interests.

