## [Peer Review File · Nature Communications]

Reviewer #1 (Remarks to the Author):

The authors introduce a new device concept, AF-MERAM, which utilises the anomalous Hall effect in a proximity-polarised Pt layer to provide a new read-out technique for a magnetoelectric AF memory. The gains made in writing threshold are striking.

I find the work to be both interesting and timely. The field of AF spintronics has garnered much interest recently and this provides another technology which should integrate well with existing schemes, and provides low energy, field-stable and non-volatile memory.

The operating temperature range of the device studied is small but the authors detail ways of improving this. On this point alone I would like a little more justification of the claim of widening the operating range by several 100K (in final few sentences of page 5). In particular I would like to see a reference or references for the final sentence of the first paragraph on page 6 which refers to intentionally lowering the anisotropy via doping.

I recommend the work for publication in Nature Communications with only minor changes.

Reviewer #2 (Remarks to the Author):

This paper reports on studies of antiferromagnetic magnetoelectric random access memory (MERAM) cells comprised of Cr₂O₃. The authors show that by removing the ferromagnetic probing layer from conventional MERAM cells, it is possible to significantly reduce the writing energy. This approach can be applied to other antiferromagnetic systems, providing possible routes to developing antiferromagnetic MERAM technology.

The paper contains large amount of information including the rich supplementary material. The results on the characterization of the Cr₂O₃ films with different buffer layers are in particular impressive and very informative. I have no question that these results should be published.

However, I am not convinced with the main message of this paper: a memory technology or logic devices can be built using the approach described. Although the authors state that the temperature window to observe robust switching of the antiferromagnetic order can be improved by material engineering, I would imagine that many readers will be confused if one knows that just a few degree Celsius changes in the temperature can disrupt the switching process (Fig. 2). It therefore sounds quite unlikely that the proposed approach will compete with the current STT-MRAM technology in the coming years. Given the broad readership of Nature communications, the current manuscript can be misleading.

In addition, the authors state the writing energy can be reduced by a factor of 50 compared to previously demonstrated MERAM with a ferromagnetic layer. However, since the authors show that the lattice mismatch can significantly influence the switching via the uncompensated moments at interfaces, I am not sure if it is fair to compare results from different groups: the structure and interfaces of the samples are likely quite different. Unless the authors provide direct comparison of the writing energy for samples with and without a magnetic layer, it may not be true that the removal of the magnetic layer is the sole cause of the 50x reduction in the writing energy.

Responses to the comments of Reviewer #1:

“The authors introduce a new device concept, AF-MERAM, which utilises the anomalous Hall effect in a proximity-polarised Pt layer to provide a new read-out technique for a magnetoelectric AF memory. The gains made in writing threshold are striking. I find the work to be both interesting and timely. The field of AF spintronics has garnered much interest recently and this provides another technology which should integrate well with existing schemes, and provides low energy, field-stable and non-volatile memory.”

“I recommend the work for publication in Nature Communications with only minor changes.”

We thank the Reviewer for these comments, which share our views of antiferromagnetic spintronics as a quickly evolving field with strong application relevance.

“The operating temperature range of the device studied is small but the authors detail ways of improving this. On this point alone I would like a little more justification of the claim of widening the operating range by several 100K (in final few sentences of page 5). In particular I would like to see a reference or references for the final sentence of the first paragraph on page 6 which refers to intentionally lowering the anisotropy via doping.”

The Néel temperature of pristine Cr₂O₃ is indeed too low to be worthwhile for commercial room-temperature applications. This issue has previously been investigated both theoretically and experimentally. We have added a new reference detailing the effects of substitutional doping on the magnetic properties of Cr₂O₃ [<https://doi.org/10.1103/PhysRevB.87.054435>], which is now Ref. 21 in the revised manuscript. While the new reference is a theoretical work, the authors' predictions regarding the effect of the Néel temperature enhancement upon boron doping have since been experimentally confirmed both qualitatively and quantitatively in Ref. 22. Therefore, we suppose that intentional cation doping would have the predicted effect as well. It should be noted that different dopants also display differences in the details of their substitutional effect. Some dopants (Ni, Co) affect nearest neighbor exchange more severely than long range exchange. Some dopants such as Mn or Fe are predicted to increase the average sublattice magnetization while the other dopants are predicted to reduce it. One should also notice, that Fe₂O₃ and Ti₂O₃ are themselves corundum structure antiferromagnets, but with different antiferromagnetic order. In particular, Fe₂O₃ possesses a Néel temperature of 950 K, but no linear magnetoelectric effect in its pristine form.

All of these facts lead us to the conclusion, that the substitutional doping of Cr₂O₃ is a complex endeavor that can yield various effects like the demonstrated increase of the Néel temperature or the change of the antiferromagnetic order type. Combining different dopants is therefore likely to address both the Néel temperature and the magnetic anisotropy, possibly to different or even inverse extents. This discussion is added as a new section I of the supplementary information. The main text at the bottom of page 5 and top of page 6 was modified. Besides the new Ref. 21, we have added a Reference to the new section I of the supplementary information. Furthermore, the statement “widened to several 100 K” has been rephrased to “widened to more than 100 K”. All modifications and extensions are highlighted in blue in the resubmitted versions of the main text and the supporting information.

Responses to the comments of Reviewer #2:

“This paper reports on studies of antiferromagnetic magnetoelectric random access memory (MERAM) cells comprised of Cr₂O₃. The authors show that by removing the ferromagnetic probing layer from conventional MERAM cells, it is possible to significantly reduce the writing energy. This approach can be applied to other antiferromagnetic systems, providing possible routes to developing antiferromagnetic MERAM technology.”

The paper contains large amount of information including the rich supplementary material. The results on the characterization of the Cr₂O₃ films with different buffer layers are in particular impressive and very informative. I have no question that these results should be published.”

We thank the Reviewer for these comments regarding the variability of the AF-MERAM concept towards other material systems.

“However, I am not convinced with the main message of this paper: a memory technology or logic devices can be built using the approach described. Although the authors state that the temperature window to observe robust switching of the antiferromagnetic order can be improved by material engineering, I would imagine that many readers will be confused if one knows that just a few degree Celsius changes in the temperature can disrupt the switching process (Fig. 2). It therefore sounds quite unlikely that the proposed approach will compete with the current STT-MRAM technology in the coming years. Given the broad readership of Nature communications, the current manuscript can be misleading.”

The temperature window is clearly an important parameter that needs to be addressed. In this respect, it should be noted that thin film Cr₂O₃-based MERAM had so far not been demonstrated to operate so close to the Néel temperature of Cr₂O₃. The operation in this extended temperature range is one of the key achievements of the concept of AF-MERAM over traditional ferromagnet-based MERAM. To enhance the temperature capabilities of magnetoelectric antiferromagnets even further, is the focus of work by other groups. In this respect, theoretical (Ref. 21) and experimental (Ref. 22) foundations have been already laid. Since we do not cover any material optimizations aimed at enhancing the Néel temperature or at reducing the intrinsic writing threshold of Cr₂O₃, we only provide the relevant references to work done in these fields. More detail on these works and on the modifications and extensions of our resubmitted manuscript can be found in our response to the respective comment of Reviewer #1.

We agree with the Reviewer that there is a strong need for future material scientific optimizations before the concept of AF-MERAM could comply with the strict requirement of commercial microelectronics. Indeed, the second part of our manuscript is devoted to the thorough analysis and identification of the current material shortcomings and we aim to provide a starting point for the optimization process by investigating the role of the gate layer. In doing so, we managed to find clear correlations between the choice of the gate layer and the degree of ferrimagnetism and the magnitude of the gate bias voltage in the thin Cr₂O₃ films. Since, we discuss the necessity for material optimization so extensively, we consider it very unlikely that readers of this article could be led to believe that the demonstrated prototype system could work in a commercial setting without further material engineering.

“In addition, the authors state the writing energy can be reduced by a factor of 50 compared to previously demonstrated MERAM with a ferromagnetic layer. However, since the authors show that the lattice mismatch can significantly influence the switching via the uncompensated moments at interfaces, I am not sure if it is fair to compare results from different groups: the structure and interfaces of the samples are likely quite different. Unless the authors provide direct comparison of the writing energy for samples with and without a magnetic layer, it may not be true that the removal of the magnetic layer is the sole cause of the 50x reduction in the writing energy.”

Although comparisons between results obtained for thin film system prepared by different groups in different chambers and different fabrications habits are often complicated, the situation is less ambiguous in this case. All of the successful demonstrations of Cr₂O₃-based thin film MERAM systems are based on the same material system, which yields consistent results throughout several groups. This system is based on (0001) cut Sapphire single crystal substrates, a roughly 20 nm thick sputtered Pt gate layer, a roughly 200 nm thick Cr₂O₃ layer prepared at about 600°C and a metallic sensing layer sputtered at room temperature. Throughout the last decade, this last sensing layer has been very thoroughly characterized. One of the key conclusions is that the stronger the exchange bias between Cr₂O₃ and a ferromagnetic Co sensing layer, the easier it is to read out the system (due to the larger loop shift) but the harder it is to write the antiferromagnet via the magnetoelectric effect. By finely tuning the exchange bias strength via intentional decoupling of Cr₂O₃ and Co, several groups eventually managed to achieve both writability of Cr₂O₃ and stable exchange bias at low temperatures. The few works which demonstrate this functionality are all included as performance references in our manuscript and are best-case scenarios selected from a much wider body of research work. Many articles published within the last decade, but also internal work done by us, showed reliable exchange bias between Cr₂O₃ and Co, reliable isolation of the gate electrode but no writability by the magnetoelectric effect because the necessary writing voltages were beyond the dielectric breakdown strength of Cr₂O₃.

On the other hand, magnetoelectric switching of pristine Cr₂O₃ has been performed since the discovery of the linear magnetoelectric effect in the early 1960ies. Many of these experiments yielded even lower values of the EH product than reported in the current manuscript as necessary to switch the AF order parameter in powder samples or to field cool to single domain states in thin film samples (Ref. 19).

Therefore, it is very likely that by selecting the best-case writing thresholds for conventional MERAM and comparing these figures to the intermediate plain Cr₂O₃ writing threshold obtained by our AF-MERAM prototype, we are in fact underestimating the reduction factor of the writing thresholds that is achieved by removing the ferromagnetic layer from a thin film MERAM system. For structurally entirely comparable MERAM and AF-MERAM systems, the reduction factor of the writing threshold could thus be even higher than 50-fold compared to the ferromagnet-containing counterparts.

An estimation of the reduction factor that could be achieved by removing the ferromagnet from the optimized traditional MERAM system can be learnt when comparing the energy contributions to the domain selection in magnetoelectric antiferromagnets (section III of the supporting information). These latter consideration yield an estimated reduction factor of the writing threshold of AF-MERAM compared to traditional MERAM on the order of 1000.

This discussion has been added as section II to the revised version of the supporting information. A reference to this new section has been added at the top of page 7 in the main text. All modifications and extensions are highlighted in blue in the resubmitted versions of the main text and the supporting information.

Reviewer #1 (Remarks to the Author):

I am happy with the changes the authors have made to the manuscript and recommend it for publication.

I have only one minor additional request. Despite being pleased with the new references and support for my query regarding the operating window, I would like to see the following sentence rephrased:

"It should be noted that this writability window can be considerably widened to more than 100 K."

I would prefer something along the lines of "It should be possible to broaden the operating window by more than 100K....."

My reason for this is that although the authors have provided justification that the upper and lower constraints can be changed, doing this together in the same heterostructure has not been demonstrated.

Reviewer #2 (Remarks to the Author):

The authors have addressed all points raised by the referees. I think the paper is suitable for publication in Nature Communications.

Response to the Reviewers comments

Reviewer #1 (Remarks to the Author):

I happy with the changes the authors have made to the manuscript and recommend it for publication. I have only one minor additional request. Despite being pleased with the new references and support for my query regarding the operating window, I would like to see the following sentence rephrased: "It should be noted that this writability window can be considerably widened to more than 100 K." I would prefer something along the lines of "It shan be possible to broaden the operating window by more than 100K....." My reason for this is that although the authors have provided justification that the upper and lower constraints can be changed, doing this together in the same heterostructure has not been demonstrated.

We agree with the remark by the Reviewer and modified the sentence accordingly to his/her suggestion.

Reviewer #2 (Remarks to the Author):

The authors have addressed all points raised by the referees. I think the paper is suitable for publication in Nature Communications.

We appreciate the positive feedback by the Reviewer.